# Comparative evaluation of sensititre YeastOne and VITEK2 antifungal susceptibility tests with CLSI broth microdilution method of clinical *Cryptococcus* isolates in Taiwan

Tzu-Ping Weng,[1,2] Shin-Wei Wang,[3] Shih-Ting Lo,[4] Shu-Li Su,[5] Ming-I Hsieh,[6] Pei-Jane Tsai,[4] Pei-Fang Tsai,[3] Chi-Jung Wu,[6] Nan-Yao Lee,[1,2,7] Wen-Chien Ko,[1,2,7] Po-Lin Chen[1,2,7]

**ABSTRACT**   Commercial antifungal susceptibility tests were available for clinical yeast isolates. However, the updated Sensititre YeastOne (SYO) version YO10C excluded *Cryptococcus* species for susceptibility testing. Uncorrelation of antifungal susceptibility patterns by SYO and therapeutic outcomes had been recently reported. We compared the performance of current commercial susceptibility tests with the standard CLSI broth microdilution (BMD) method for clinical *Cryptococcus* isolates. Forty-seven clinical *Cryptococcus* isolates were included from 1 January 2012 to 30 June 2023, among which 44 isolates were *Cryptococcus neoformans* while 3 were *Cryptococcus gattii*. The performance of SYO version YO10C and VITEK2 YS09 was compared with the CLSI BMD method and correlated with MLST analysis and ERG11 mutation detection. Non-wild-type (non-WT) strains to amphotericin B (AMB) were observed in 11 isolates with the CLSI BMD method and 8 with SYO among 44 *C. neoformans* isolates, but only 1 isolate was classified as non-WT by both methods. Additionally, all *C. neoformans* isolates were susceptible to AMB with their MIC ≤1 µg/mL according to the clinical breakpoint defined by EUCAST. Non-WT to FLC were observed in 5 *C. neoformans* isolates with SYO, but they were classified as WT by CLSI BMD and VITEK2. The essential agreements between SYO and CLSI BMD were >90% to most antifungal agents except ITC in *C. neoformans* isolates (64%) and AMB in *C. gattii* group (67%). Between SYO and CLSI BMD, the major error (ME) rates were 11% (*n* = 5) to FLC, 5% (*n* = 2) to ITC, and 2% (*n* = 1) to 5FC in *C. neoformans* isolates, and the very major error to 5FC was found in one *C. gattii* isolate. ERG11 mutation with identical I199V was detected in 89% (*n* = 39) *C. neoformans* isolates, and 97% (*n* = 38) of them belonged to sequence type (ST) 5. The ERG11 mutation or cryptococcal ST was not associated with a decrease of antifungal susceptibilities. ME of FLC by SYO version YO10C compared to the CLSI BMD method reached up to 11% of *C. neoformans* isolates. The results of FLC MIC by SYO should be interpreted cautiously and correlated with therapeutic response, and further verification with the CLSI BMD method or VITEK2 is required.

**IMPORTANCE**   The study pointed out the major errors of fluconazole susceptibility results in clinical *Cryptococcus neoformans* isolates between the commercial Sensititre YeastOne Susceptibility Plate version YO10C and the standard CLSI broth microdilution method. The results should be interpreted carefully with clinical correlation, and a different method of antifungal susceptibility testing should be considered if a discrepancy of susceptibility results is suspected.

**KEYWORDS**   *Cryptococcus* infection, antifungal susceptibility, MLST, ERG11 mutation

Address correspondence to Po-Lin Chen, cplin@mail.ncku.edu.tw.

The authors declare no conflict of interest.

*C*ryptococcus is a genus of yeast-like fungi causing infections in human beings, composed of two species—*Cryptococcus neoformans* and *Cryptococcus gattii*. The mortality and complication rates remain high in patients with cryptococcosis despite evolutions of diagnostic tools and relevant experiences of disease managements (1–5).

The immunocompromised individuals, such as those with hematologic malignancies, recipients of solid organ transplants with ongoing immunosuppressive therapy, persons with long-term glucocorticoid therapy, or people living with HIV with CD4$^+$ T lymphocyte counts of <200 /µL, were at high risk to be infected with *C. neoformans*, which was mainly found in soils contaminated with avian excreta or humid soils contaminated with pigeon droppings (6, 7). *C. gattii* inhabits a variety of arboreal species, including several types of eucalyptus trees. The *C. gattii*-related disease is not associated with specific immune deficits and often occurred in immunocompetent individuals (6). Patients infected with *C. gattii* have more evident environmental exposure and higher humoral response (8).

The *in vitro* resistance of *C. neoformans* isolates to antifungals was uncommon before 2000 (9). However, approximately 25% of *C. neoformans* isolates in North America displayed minimum inhibitory concentration (MIC) to fluconazole (FLC) >8 µg/mL in a global surveillance program during 1990–2004 (10). Besides, the *C. neoformans* isolates from HIV patients in Uganda presented elevated FLC MIC during 2010–2014 compared with the previous study period during 1998–1999 (11). The mean prevalence of FLC resistance reached 12.1% among 4,995 *Cryptococcus* isolates from 3,210 patients from 1988 to May 2017 in a systemic review though the MIC cutoff defined for FLC resistance was different between studies (12).

FLC preferentially binds to and inhibits lanosterol 14-α demethylase (ERG 11), a fungal cytochrome P450-linked mono-oxygenase, obstructing the conversion of fungal lanosterol to ergosterol, thereby inhibiting membrane sterol synthesis and preventing fungal cell replication (13). Elevated MIC to FLC was observed in *C. neoformans* isolates with alterations of ERG11 gene, and specific mutations of Y145F, G484S, G470R, and I99V had been identified in clinical *Cryptococcus* isolates (14–16).

It is important to guide antifungal therapy according to antifungal MIC results. However, a proportion of patients experienced disease relapse with elevated FLC MIC of *Cryptococcus* isolates despite FLC was recommended as the primary regimen for consolidation and maintenance therapies for cryptococcosis according to the clinical practice guidelines for the management of cryptococcal disease by the Infectious Diseases Society of America (IDSA) (17).

There are several commercial kits available to determine antifungal susceptibility results. They are simple, effective, standardized, and relatively uncomplicated to implement in clinical microbiological laboratories compared with the standard CLSI BMD method, which was relatively costly, time-consuming, and technically complicated for laboratory personnel. The Vitek 2 Susceptibility Card YS09 (bioMe'rieux, North Carolina, USA) for VITEK 2 Systems included *C. neoformans* isolates intended for use. However, the manufacturer noted the limitations in susceptibility results of *C. neoformans* to FLC and suggested an alternative testing method for FLC.

The Sensititre YeastOne (SYO) Susceptibility Plate (Thermo Scientific, West Sussex, United Kingdom), which was a commercial *in vitro* broth microdilution (BMD) plate, provided quantitative MIC measurements for non-fastidious yeasts. It was mainly used for the clinical *Candida* isolates for its well correlation with the CLSI BMD method (18–20), and the performance had been proved for the clinical *Cryptococcus* isolates (18, 21, 22).

However, lack of association between antifungal susceptibilities by SYO and clinical outcomes of patients with cryptococcal meningitis and fungemia had been recently reported (23, 24). The antifungal susceptibility patterns by SYO did not correlate with therapeutic outcomes in patients with cryptococcosis.

The version of the SYO used for antifungal susceptibility testing of *Cryptococcus* isolates was not reported in most previous studies (18–21). According to the manufacturer's instructions, the SYO version YO10 (25) provided *in vitro* susceptibility testing for non-fastidious yeasts including *Candida* species, *Cryptococcus* species, *Aspergillus*

species, and miscellaneous rapid growing yeast species. However, the *Cryptococcus* species was no more claimed by the manufacturer intended for susceptibility testing in the updated SYO version YO10C (26). The investigations about the performance of SYO YO10C in *Cryptococcus* species are lacking, so it should be evaluated for off-label use in clinical *Cryptococcus* isolates.

For *Cryptococcus* strains with elevated antifungal MICs, there had been no approval of therapeutic antifungal agent by the US Food and Drug Administration (FDA) for the past 20 years. Fosmanogepix, an *N*-phosphonooxymethylene prodrug of manogepix (MGX), was an inhibitor of the fungal enzyme Gwt1, and it was granted Fast Track designations by the US FDA for the treatment of invasive candidiasis, invasive aspergillosis, scedospor-iosis, fusariosis, mucormycosis, cryptococcosis, and coccidioidomycosis (27). The MGX had been proved to demonstrate broad spectrum *in vitro* activity against yeasts and molds in phase 2 clinical trials (28), and the activities of the MGX analogs were proved against several isolates of *C. neoformans* and *C. gattii* (29). However, more *in vitro* studies with *Cryptococcus* isolates are required.

The aim of our study was to investigate the antifungal susceptibility spectrum of amphotericin B (AMB), 5-flucytosine (5FC), FLC, voriconazole (VRC), itraconazole (ITC), Posaconazole (PSC), and isavuconazole (ISC) between the standard CLSI BMD method with the SYO version YO10C and VITEK2 YS09. The susceptibilities of *Cryptococcus* isolates to MGX were evaluated with CLSI BMD. The phylogenetic analysis of multi-locus sequence typing (MLST) and the ERG11 (lanosterol 14-α demethylase) mutation responsible for possible FLC resistance were investigated for the included *Cryptococcus* isolates.

## MATERIALS AND METHODS

### Study design and isolates collection

The study was approved by the Institutional Review Board (B-ER-111-284) of National Cheng Kung University Hospital (NCKUH) in southern Taiwan. The *Cryptococcus* isolates stored at the Department of Pathology and the Centers for Infection Control of NCKUH from 1 January 2012 to 30 June 2023 were included. The included isolates were routinely identified to species level with the assistance of matrix-assisted laser desorp-tion ionization–time of flight–mass spectrometry (MALDI-TOF-MS) (bioMe´rieux, Marcy L'Etoile, France) (30). The repeated isolates from the same patient or the isolate collected without aseptic procedure or not from sterile body fluid or tissue were excluded. The stored clinical *Cryptococcus* isolates were stored at −80°C in 20% glycerol (VWR, Solon, USA) until antifungal susceptibility testing for study.

### Antifungal susceptibility testing

The susceptibility testing of the standard BMD method was according to the CLSI M27 reference method for broth dilution antifungal susceptibility testing of yeasts (31). The stock solutions were prepared in water or other solvents according to CLSI M60 (32). The antifungal agent concentrations were ranged 0.03125 to 16 µg/mL for AMB (Sigma-Aldrich, Saint Louis, Missouri, USA), 0.125 to 64 µg/mL for 5FC (Sigma-Aldrich, Saint Louis, Missouri, USA), 0.125 to 64 µg/mL for FLC (Sigma-Aldrich, Saint Louis, Missouri, USA), 0.03125 to 16 µg/mL for VRC (Sigma-Aldrich, Saint Louis, Missouri, USA), ITC (Sigma-Aldrich, Saint Louis, Missouri, USA), PSA (Sigma-Aldrich, Saint Louis, Missouri, USA), ISC (Sigma-Aldrich, Saint Louis, Missouri, USA), and the novel antifungal regimen of MGX (MedChemExpress, Monmouth, New Jersey, USA). The antifungal agent was dissolved in Roswell Park Memorial Institute (RPMI) 1640 and buffered to pH 7 ± 0.1 at 25°C with MOPS [3-(*N*-morpholino) propanesulfonic acid]. The 96-well microplates were stored at −70°C until use. The stored isolates were subcultured on Sabouraud dextrose agar at 35°C for 24–48 h. The colonies were suspended in saline and were diluted until $0.5 \times 10^3$ to $2.5 \times 10^3$ CFU/mL. The 96-well microdilution plates were incubated at 35°C

at ambient air for 72 h. The *C. parapsilosis* ATCC 22019TM and *C. krusei* ATCC 6258TM from the American Type Culture Collection (ATCC) were used for each BMD method testing for quality control (32). The turbidity of wells was scored under normal laboratory lighting with a manual mirror compared with the growth control well. The MIC for AMB was defined as the lowest drug concentration in which there was no growth (optically clear). For 5FC, FLC, VRC, ITA, PSA, and ISC, the MIC was defined as the lowest concentration at which there was a 50% decrease in growth (prominent decrease in turbidity). Due to the endpoint of MGX not being well defined and significant variability, a 50% reduction in growth relative to the antifungal agent-free growth control was determined as the MIC, following the CLSI's recommendation to improve interlaboratory agreement and reproducibility (31).

The SYO YO10C and VITEK2 YS09 susceptibility testing was performed according to the manufacturers' instructions, and the *C. parapsilosis* ATCC 22019TM and *C. krusei* ATCC 6258TM from the American Type Culture Collection (ATCC) were processed for quality control of the two MIC systems. The SYO YO10C was a colorimetric microdilution plate with different-dosed antifungal agents at appropriate dilutions and a colorimetric indicator. The organism concentration should reach $1.5-8 \times 10^3$ CFU/mL after inoculation, and the *Cryptococcus* isolates were incubated at 35°C for 72 h. The MIC of AMB ranged 0.12 to 8 µg/mL, 5FC ranged 0.06 to 64 µg/mL, FLC ranged 0.12 to 256 µg/mL, ITC ranged 0.015 to 16 µg/mL, VRC and PSC ranged 0.008 to 8 µg/mL were tested. The MIC of AMB was defined as the lowest drug concentration that prevented any discernible color change. For 5FC and azoles, the MIC was read as the first well showing a less intense color change compared to the positive growth control well.

The VITEK2 YS09 was an automated test methodology of the doubling dilution technique for antifungal MIC determined by the microdilution method. The organism suspension to be tested was diluted to the standardized concentration in 0.45% saline before being used to rehydrate the antimicrobial medium within the card. The card was filled, sealed, and placed into the instrument incubator and reader. Growth of each well in card was monitored over a period of time which was up to 36 h for yeasts. At the completion of the incubation cycle, the MIC values of 5FC ranged 1 to 64 µg/mL, FLC ranged 0.5 to 64 µg/mL, and VRC ranged 0.12 to 8 µg/mL were determined for each antimicrobial contained in the card.

## MIC interpretations

The interpretations of measured MIC were according to the epidemiological cutoff value (ECV) (33, 34) according to the criteria in CLSI M59 for antifungal susceptibility testing (35). The ECV was used to distinguish wild-type (WT) isolates and non-WT isolates possibly harboring acquired resistance mechanisms. It could not classify the isolate which was treatable (susceptible) or non-treatable (resistant) or predict therapeutic response. For *C. neoformans*, the ECV to AMB was 0.5 µg/mL, 8 µg/mL for 5FC and FLC, and 0.25 µg/mL for VRC, ITC, and PSC. For *C. gattii*, the ECV for AMB was defined 0.5 µg/mL, 4 µg/mL for 5FC, 16 µg/mL for FLC, and 0.5 µg/mL for VRC and ITC.

According to the EUCAST antifungal ECOFFs and clinical breakpoints for yeasts, molds, and dermatophytes (Version 4.0) (36), the clinical breakpoint to AMB was defined at 1 µg/mL for *C. neoformans*. For *C. neoformans*, the ECV to VRC and PSC were defined at 0.5 µg/mL. For *C. gattii*, the ECV to AMB was defined at 0.5 µg/mL, and the ECV to PSC was defined at 1 µg/mL (Table S1).

## MLST and phylogenetic analysis

Seven MLST loci of CAP59, GPD1, LAC1, PLB1, SOD1, URA5, and IGS1 for the *C. neoformans* and *C. gattii* isolates were amplified according to the consensus of the International Society of Human and Animal Mycology (ISHAM) (37), except for the primers used to amplify the genes GPD1 and LAC1 of *C. neoformans*. The GPD1 locus of *C. neoformans* was amplified with the primers of GPD1cn-F 59ATGGTCGTCAAGGTTGGAAT 39 and GPD1cn-R 59 GTATTCGGCACCAGCCTCA 39, and the LAC1 locus of *C. neoformans* was

amplified with the primers LAC1cn-F 59 GGCGATACTATTATCGTA39 and LAC1cn-R 59-TTC TGGAGTGGCTAGAGC39 (38). The allele type and sequences type (ST) were assigned by sequence comparison for the *Cryptococcus* isolates in the Fungal MLST Database (37).

## Detection of ERG11 gene mutation

The *Cryptococcus* colonies were suspended in saline, and the genomic DNA was extracted with the DNeasy Blood & Tissue kit (Qiagen, Hilden, Germany) according to the manufacturer's instructions. Amplification of the ERG11 gene by PCR was initiated with denaturation at 94℃ for 5 min, 35 cycles of denaturation at 94℃ for 1 min, annealing at 57℃ for 30 s, and extension at 72℃ for 1 min, with an extension cycle at 72℃ for 10 min. PCR products were purified and sequenced using a model 3730XL sequencing system (Applied Biosystems, Taipei, Taiwan), and the genomic sequences were aligned and translated to amino acids with DS GENE 1.5 (Accerlrys Inc.). The reference strain of *C. neoformans* var. *grubii* CM64 (GenBank accession no. APKS01000005) was used to annotate the obtained sequences and to identify single-nucleotide polymorphisms (SNPs) and amino acid substitutions.

## Definition

An ECV was defined as the MIC value that located at the upper limit of the WT *Cryptococcus* isolate. A non-WT isolate was classified as its MIC greater than the ECV to a specific antifungal regimen. The category agreement was defined the agreement of results between the MIC testing result and the reference result. The essential agreement was the antifungal susceptibility results within two doubling dilution steps from the MIC value established with the reference method. A very major error (VME) was defined as an instance where isolates were classified as non-wild type (non-WT) by the standard CLSI BMD method but were identified as wild type (WT) using the commercial system. A major error (ME) occurred when the opposite classification results happened (39).

## RESULTS

### Baseline characteristics

Of the 48 stored *Cryptococcus* isolates at NCKUH, one of the two isolates was excluded for its origin from the same patient. Among the included 47 *Cryptococcus* isolates, 44 isolates were identified as *C. neoformans,* while 3 isolates were *C. gattii*. Of the 44 isolates of *C. neoformans*, 31 were collected from January 2012 to December 2016, and the rest 13 were stored from January 2017 to June 2023. Three isolates of *C. gattii* were collected from October 2021 to February 2023.

### Distributions of *in vitro* antifungal susceptibilities

All of *Cryptococcus* isolates grew well in the microdilution wells in 72 h, and the isolates grew well in the SYO and VITEK2 plates as well. Of the 44 *C. neoformans* isolates, the distribution of AMB MIC was narrower (0.5–1 µg/mL) with the CLSI BMD method compared to SYO (<0.12–1 µg/mL). The MIC > ECV to AMB was observed in 11 isolates with the CLSI BMD method, and 8 with SYO according to the ECV defined by CLSI M59 (35) (Table 1). However, there was only one isolate classified as non-WT using both CLSI BMD and SYO (Tables 2 and 3). Additionally, all of the 44 *C. neoformans* isolates were classified to be susceptible to AMB according to the clinical breakpoint defined by EUCAST (36) (Table S1).

For susceptibilities to 5FC, the distributions of MIC were narrower with CLSI BMD (2–8 µg/mL) and VITEK2 (≤1–2 µg/mL) compared to SYO (2–16 µg/mL), and one non-WT strain was observed with SYO. For susceptibilities to FLC, there was no non-WT strain observed with CLSI BMD or VITEK2, but five isolates were classified as non-WT by SYO. Among the five isolates belonged to non-WT to FLC by SYO, there was only one isolate classified as non-WT to AMB (Table 3). For VRC susceptibilities, the MIC determined by

**TABLE 1** *In vitro* antifungal susceptibilities of 44 *C. neoformans* isolates determined by the CLSI broth microdilution method, Sensititre YeastOne, and VITEK2 Susceptibility Card[a]

| Drug Method | MIC (µg/mL) <0.03 | 0.03 | 0.06 | 0.12 | 0.25 | 0.5 | 1 | 2 | 4 | 8 | 16 | 32 | Non-WT (CLSI) | Non-WT (EUCAST) |
|---|---|---|---|---|---|---|---|---|---|---|---|---|---|---|
| **AMB** | | | | | | | | | | | | | | |
| BMD | | | | | | 33 | 11 | | | | | | 11 | 0 |
| SYO | | | 1 (<0.12) | | 2 | 33 | 8 | | | | | | 8 | 0 |
| **5FC** | | | | | | | | | | | | | | |
| BMD | | | | | | | | 2 | 20 | 22 | | | 0 | |
| SYO | | | | | | | | 10 | 29 | 4 | 1 | | 1 | |
| VITEK2 | | | | | | | 17 (≤1) | 27 | | | | | 0 | |
| **FLC** | | | | | | | | | | | | | | |
| BMD | | | | | | 1 | 1 | 8 | 28 | 6 | | | 0 | |
| SYO | | | | | | | | 2 | 4 | 33 | 3 | 2 | 5 | |
| VITEK2 | | | | | | 1 (≤0.5) | 3 | 40 | | | | | 0 | |
| **VRC** | | | | | | | | | | | | | | |
| BMD | 2 | 13 | 25 | 4 | | | | | | | | | 0 | 0 |
| SYO | 1 (0.015) | 1 | 22 | 18 | 2 | | | | | | | | 0 | 0 |
| VITEK2 | | | | 43 (≤0.12)[b] | | | | | | | | | 0 | 0 |
| **ITC** | | | | | | | | | | | | | | |
| BMD | 12 | 9 | 16 | 7 | | | | | | | | | 0 | |
| SYO | | 2 | | 26 | 14 | 2 | | | | | | | 2 | |
| **PSC** | | | | | | | | | | | | | | |
| BMD | | 1 | 3 | 30 | 10 | | | | | | | | 0 | 0 |
| SYO | | | | 8 | 33 | 3 | | | | | | | 3 | 0 |
| **ISC** | | | | | | | | | | | | | | |
| BMD | 2 | 13 | 21 | 6 | 2 | | | | | | | | | |
| **MGX** | | | | | | | | | | | | | | |
| BMD | | | | 2 | 1 | | 8 | 24 | 8 | 1 | | | | |

[a]CLSI, Clinical and Laboratory Standards Institute; WT, wild type; EUCAST, European Committee on Antimicrobial Susceptibility Testing; BMD, broth microdilution; SYO, Sensititre YeastOne; AMB, amphotericin B; 5FC, flucytosine; FLC, fluconazole; VRC, voriconazole; ITC, itraconazole; PSC, posaconazole; ISC, isavuconazole; MGX, manogepix.
[b]MIC to VRC was undetermined in one isolate by VITEK2 due to insufficient growth in positive control well despite repeated testing for three times.

VITEK2 of 43 *C. neoformans* isolates was ≤0.12 µg/mL, and the MIC of 1 isolate was undetermined due to insufficient growth in positive control well despite repeated testing for three times. Overall, the MICs to ITC and PSC were all ≤0.5 µg/mL determined by BMD and SYO, while the MICs to ISC were all ≤0.5 µg/mL by BMD. The PSC MIC of three *C. neoformans* isolates was 0.5 µg/mL by SYO, which was classified as non-WT by CLSI M69, but three of them were classified as WT by EUCAST (36). Among three *C. gattii* isolates, one was classified as non-WT to 5FC with the CLSI BMD method, but it was classified as WT by both SYO and VITEK2. The MIC distributions of MGX ranged from 0.12 to 8 µg/mL among 47 *Cryptococcus* isolates (Tables 1 and 4).

## Agreements and errors between three methods

The essential agreement between the CLSI BMD method and SYO for most antifungal agents was >90% except for ITC (64%) in *C. neoformans* and AMB (67%) in *C. gattii* isolates. The agreement within one twofold dilution for most drugs was >80% except for FLC (75%) and ITC (45%) in *C. neoformans*, and 5FC (67%) in *C. gattii* group (Table 5).

The VME to AMB between CLSI BMD and SYO were found in 23% (*n* = 10) of *C. neoformans* isolates, while the ME was observed in 16% (*n* = 7) according to the CLSI M59 (35). However, there was no VME or ME between the CLSI BMD method and SYO according to the AMB clinical breakpoint defined by EUCAST (36). The ME was observed in 11% (*n* = 5) *C. neoformans* isolates to FLC, 5% (*n* = 2) to ITC, and 2% (*n* = 1) to 5FC between the two methods. Among three *C. gattii* isolates, there was only one VME to 5FC between CLSI BMD and SYO (Table 5; Table S1).

**TABLE 2** *In vitro* antifungal susceptibilities of 44 *C. neoformans* isolates determined by the CLSI broth microdilution method[a,b]

| Isolate number | Antifungal susceptibilities (MIC), µg/mL | | | | | | | |
|---|---|---|---|---|---|---|---|---|
| | AMB | 5FC | FLC | VRC | ITC | PSC | ISC | MGX |
| 1 | **1** | 8 | 4 | 0.03 | 0.12 | 0.12 | 0.03 | 1 |
| 2 | **1** | 4 | 4 | 0.03 | 0.03 | 0.12 | 0.06 | 4 |
| 3 | 0.5 | 4 | 8 | 0.12 | 0.12 | 0.12 | 0.12 | 2 |
| 4 | 0.5 | 8 | 4 | 0.03 | 0.06 | 0.12 | 0.06 | 2 |
| 5 | 0.5 | 4 | 4 | 0.03 | 0.03 | 0.12 | 0.06 | 2 |
| 6 | 0.5 | 4 | 4 | 0.06 | 0.12 | 0.12 | 0.12 | 4 |
| 7 | 0.5 | 4 | 4 | 0.06 | 0.06 | 0.12 | 0.12 | 2 |
| 8 | **1** | 8 | 4 | 0.06 | 0.06 | 0.12 | 0.06 | 2 |
| 9 | **1** | 8 | 4 | 0.06 | 0.06 | 0.12 | 0.06 | 4 |
| 10 | **1** | 2 | 2 | 0.03 | 0.03 | 0.06 | 0.03 | 2 |
| 11 | **1** | 4 | 4 | 0.06 | 0.03 | 0.12 | 0.03 | 4 |
| 12 | **1** | 4 | 4 | 0.06 | 0.03 | 0.12 | 0.06 | 1 |
| 13 | **1** | 4 | 0.5 | 0.03 | 0.03 | 0.03 | 0.03 | 0.25 |
| 14 | **1** | 4 | 2 | 0.03 | 0.03 | 0.12 | 0.03 | 2 |
| 15 | 0.5 | 8 | 4 | 0.06 | 0.12 | 0.25 | 0.06 | 2 |
| 16 | 0.5 | 8 | 8 | 0.06 | 0.06 | 0.25 | 0.06 | 2 |
| 17 | 0.5 | 4 | 4 | 0.06 | 0.12 | 0.25 | 0.12 | 2 |
| 18 | 0.5 | 4 | 8 | 0.03 | <0.03 | 0.12 | 0.03 | 1 |
| 19 | 0.5 | 4 | 2 | 0.06 | <0.03 | 0.25 | 0.06 | 2 |
| 20 | 0.5 | 8 | 2 | 0.06 | <0.03 | 0.12 | 0.06 | 2 |
| 21 | 0.5 | 4 | 2 | 0.12 | 0.06 | 0.25 | 0.06 | 1 |
| 22 | 0.5 | 2 | 4 | 0.03 | <0.03 | 0.12 | 0.03 | 1 |
| 23 | 0.5 | 4 | 4 | 0.06 | <0.03 | 0.12 | 0.03 | 8 |
| 24 | 0.5 | 4 | 4 | 0.06 | <0.03 | 0.12 | 0.06 | 4 |
| 25 | 0.5 | 8 | 4 | 0.06 | 0.06 | 0.12 | 0.06 | 1 |
| 26 | 0.5 | 4 | 4 | 0.06 | 0.06 | 0.25 | 0.06 | 2 |
| 27 | 0.5 | 8 | 4 | 0.06 | 0.06 | 0.12 | 0.06 | 2 |
| 28 | 0.5 | 8 | 8 | 0.12 | 0.12 | 0.25 | 0.25 | 4 |
| 29 | **1** | 8 | 4 | 0.06 | 0.06 | 0.12 | 0.12 | 2 |
| 30 | **1** | 4 | 8 | 0.12 | 0.12 | 0.25 | 0.25 | 4 |
| 31 | 0.5 | 8 | 4 | 0.06 | 0.06 | 0.12 | 0.06 | 2 |
| 32 | 0.5 | 8 | 4 | 0.06 | 0.06 | 0.12 | 0.06 | 2 |
| 33 | 0.5 | 8 | 1 | <0.03 | <0.03 | 0.06 | <0.03 | 0.12 |
| 34 | 0.5 | 4 | 2 | <0.03 | <0.03 | 0.12 | 0.03 | 2 |
| 35 | 0.5 | 4 | 2 | 0.06 | <0.03 | 0.06 | 0.03 | 0.12 |
| 36 | 0.5 | 4 | 4 | 0.03 | 0.06 | 0.25 | 0.03 | 4 |
| 37 | 0.5 | 8 | 4 | 0.03 | 0.03 | 0.12 | 0.06 | 2 |
| 38 | 0.5 | 8 | 2 | 0.03 | <0.03 | 0.12 | <0.03 | 2 |
| 39 | 0.5 | 8 | 4 | 0.06 | <0.03 | 0.12 | 0.06 | 1 |
| 40 | 0.5 | 8 | 4 | 0.06 | 0.06 | 0.12 | 0.03 | 2 |
| 41 | 0.5 | 8 | 4 | 0.06 | 0.06 | 0.12 | 0.06 | 1 |
| 42 | 0.5 | 8 | 4 | 0.03 | 0.03 | 0.12 | 0.03 | 2 |
| 43 | 0.5 | 8 | 8 | 0.06 | 0.06 | 0.25 | 0.12 | 2 |
| 44 | 0.5 | 8 | 4 | 0.06 | <0.03 | 0.12 | 0.06 | 2 |

[a]Non-WT according to ECV by CLSI indicated in boldfaced number.
[b]CLSI, Clinical and Laboratory Standards Institute; AMB, amphotericin B; 5FC, flucytosine; FLC, fluconazole; VRC, voriconazole; ITC, itraconazole; PSC, posaconazole; ISC, isavuconazole; MGX, manogepix; WT, wild type; ECV, epidemiological cutoff value.

The essential agreement between CLSI BMD and VITEK2 for 44 *C. neoformans* isolates reached 98% for FLC, and the agreement of one twofold dilution for FLC maintained up to 84%. The essential agreement for 5FC was 61% between the two methods, and

**TABLE 3** *In vitro* antifungal susceptibilities of 44 *C. neoformans* isolates determined by Sensititre YeastOne[a,b]

| Isolate number | Antifungal susceptibilities (MIC), µg/mL | | | | | |
|---|---|---|---|---|---|---|
| | AMB | 5FC | FLC | VRC | ITC | PSC |
| 1 | 0.5 | 4 | 8 | 0.12 | 0.25 | 0.25 |
| 2 | 0.5 | 4 | 4 | 0.06 | 0.12 | 0.12 |
| 3 | 0.5 | 4 | **16** | 0.12 | 0.25 | 0.25 |
| 4 | 0.5 | 4 | 8 | 0.06 | 0.12 | 0.25 |
| 5 | 0.5 | 2 | 8 | 0.06 | 0.12 | 0.25 |
| 6 | **1** | 2 | 8 | 0.06 | 0.12 | 0.25 |
| 7 | 0.5 | 4 | 8 | 0.06 | 0.12 | 0.25 |
| 8 | 0.5 | 4 | 8 | 0.12 | 0.12 | 0.25 |
| 9 | 0.5 | 4 | 8 | 0.12 | 0.12 | 0.25 |
| 10 | 0.25 | 2 | 8 | 0.12 | 0.25 | 0.25 |
| 11 | 0.5 | 4 | 8 | 0.12 | 0.12 | 0.25 |
| 12 | 0.5 | 8 | **32** | 0.25 | 0.25 | **0.5** |
| 13 | 0.5 | 4 | 2 | 0.015 | 0.03 | 0.12 |
| 14 | 0.5 | 2 | 8 | 0.06 | **0.5** | 0.25 |
| 15 | 0.5 | 4 | 8 | 0.06 | 0.12 | 0.25 |
| 16 | **1** | 8 | 8 | 0.12 | 0.25 | 0.25 |
| 17 | 0.5 | 4 | **16** | 0.12 | 0.25 | **0.5** |
| 18 | 0.5 | 4 | 8 | 0.12 | 0.25 | 0.25 |
| 19 | 0.5 | 4 | 8 | 0.12 | 0.25 | 0.25 |
| 20 | 0.5 | 2 | 4 | 0.06 | 0.12 | 0.12 |
| 21 | 0.5 | 2 | **16** | 0.12 | 0.12 | 0.25 |
| 22 | 0.5 | 2 | 8 | 0.12 | 0.12 | 0.25 |
| 23 | 0.5 | 4 | 8 | 0.12 | 0.25 | 0.25 |
| 24 | 0.5 | 4 | 8 | 0.12 | 0.25 | 0.25 |
| 25 | 0.5 | 4 | 8 | 0.06 | 0.12 | 0.25 |
| 26 | **1** | 2 | 8 | 0.12 | 0.25 | 0.25 |
| 27 | 0.5 | 4 | 8 | 0.06 | 0.12 | 0.25 |
| 28 | **1** | 8 | **32** | 0.25 | **0.5** | **0.5** |
| 29 | **1** | 4 | 8 | 0.06 | 0.12 | 0.25 |
| 30 | 0.5 | 2 | 8 | 0.06 | 0.12 | 0.25 |
| 31 | 0.5 | 4 | 8 | 0.06 | 0.12 | 0.12 |
| 32 | 0.25 | 2 | 8 | 0.06 | 0.12 | 0.25 |
| 33 | **1** | 4 | 4 | 0.06 | 0.12 | 0.25 |
| 34 | **1** | 4 | 8 | 0.12 | 0.25 | 0.25 |
| 35 | 0.5 | 8 | 2 | 0.03 | 0.03 | 0.12 |
| 36 | 0.5 | 4 | 8 | 0.06 | 0.12 | 0.25 |
| 37 | 0.5 | 4 | 8 | 0.06 | 0.12 | 0.12 |
| 38 | 0.5 | 4 | 8 | 0.06 | 0.12 | 0.25 |
| 39 | 0.5 | 4 | 8 | 0.06 | 0.12 | 0.25 |
| 40 | 0.5 | 4 | 8 | 0.06 | 0.12 | 0.25 |
| 41 | 0.5 | 4 | 4 | 0.06 | 0.12 | 0.12 |
| 42 | <0.12 | 4 | 8 | 0.12 | 0.25 | 0.25 |
| 43 | 0.5 | 4 | 8 | 0.12 | 0.25 | 0.25 |
| 44 | **1** | **16** | 8 | 0.06 | 0.12 | 0.12 |

[a]Non-WT according to ECV by CLSI indicated in boldfaced number.
[b]AMB, amphotericin B; 5FC, flucytosine; FLC, fluconazole; VRC, voriconazole; ITC, itraconazole; PSC, posaconazole; WT, wild type; ECV, epidemiological cutoff value; CLSI, Clinical and Laboratory Standards Institute.

the MIC distributions of 5FC with the CLSI BMD method were 2–8 µg/mL, while VITEK2 were ≤1–2 µg/mL. The agreement was not applicable for VRC between the two methods because the MIC to VRC of all *Cryptococcus* isolates were ≤0.12 µg/mL except one *C. neoformans* isolate with undetermined result. Among three *C. gattii* isolates, the essential

**TABLE 4** *In vitro* antifungal susceptibilities of three *Cryptococcus gattii* isolates determined by the CLSI broth microdilution method, Sensititre YeastOne, and VITEK2 Susceptibility Card[a]

| Drug | MIC (µg/mL) | | | | | | | | | Non-WT | Non-WT |
|---|---|---|---|---|---|---|---|---|---|---|---|
| Method | 0.06 | 0.12 | 0.25 | 0.5 | 1 | 2 | 4 | 8 | 16 | (CLSI) | (EUCAST) |
| AMB | | | | | | | | | | | |
| BMD | | | | 3 | | | | | | 0 | 0 |
| SYO | | | | 3 | | | | | | 0 | 0 |
| 5FC | | | | | | | | | | | |
| BMD | | | | | | 2 | 1 | | | 1 | |
| SYO | | | | | | 3 | | | | 0 | |
| VITEK2 | | | | | 3 (≤1) | | | | | 0 | |
| FLC | | | | | | | | | | | |
| BMD | | | | | | | | 2 | 1 | 0 | |
| SYO | | | | | | | | 1 | 2 | 0 | |
| VITEK2 | | | | | | 3 | | | | 0 | |
| VRC | | | | | | | | | | | |
| BMD | | 1 | 2 | | | | | | | 0 | |
| SYO | 1 | | 2 | | | | | | | 0 | |
| VITEK2 | | 3 (≤0.12) | | | | | | | | 0 | |
| ITC | | | | | | | | | | | |
| BMD | | | 3 | | | | | | | 0 | |
| SYO | | 2 | 1 | | | | | | | 0 | |
| PSC | | | | | | | | | | | |
| BMD | | | 3 | | | | | | | | 0 |
| SYO | | 1 | 2 | | | | | | | | 0 |
| ISC | | | | | | | | | | | |
| BMD | | 1 | 2 | | | | | | | | |
| MGX | | | | | | | | | | | |
| BMD | | | | | | 3 | | | | | |

[a]CLSI, Clinical and Laboratory Standards Institute; WT, wild type; EUCAST, European Committee on Antimicrobial Susceptibility Testing; BMD, broth microdilution; SYO, Sensititre YeastOne; AMB, amphotericin B; 5FC, flucytosine; FLC, fluconazole; VRC, voriconazole; ITC, itraconazole; PSC, posaconazole; ISC, isavuconazole; MGX, manogepix.

agreement for 5FC and FLC was both below 70%, and one VME was observed between CLSI BMD and VITEK2 (Table 6).

## MLST and sequence types

Among the 44 *C. neoformans* isolates, 86% (*n* = 38) of isolates was classified as ST 5, while the other 6 isolates belonged to ST 2, ST 4, ST 6, ST 31, ST 32, and ST 93, respectively. There was no difference of antifungal susceptibility distributions in *C. neoformans* isolates belonged to ST 5 or not (Table S2).

Among the 3 *C. gattii* isolates, one isolate belonged to ST 328, and two isolates were confirmed as a new ST 579 with identical seven loci including the new allele type of SOD1 137 after submitting the sequences to MLST database for confirmation.

## ERG11 mutations

Among the 44 *C. neoformans* isolates, the ERG11 mutation with identical I199V was detected in 89% (*n* = 39) of isolates. Interestingly, 97% (*n* = 38) isolates matched to MLST ST 5 except one isolate belonged to ST 2. The MIC to FLC of the 39 *C. neoformans* isolates was ≤8 µg/mL with the CLSI BMD method. No difference of antifungal susceptibility distributions was observed in the *C. neoformans* isolates with and without ERG11 mutations (Table S3).

**TABLE 5** Agreements and errors of 47 *Cryptococcus* isolates between the CLSI broth microdilution method and Sensititre YeastOne[a]

| Species | Drug | Method | Agreement (%) | | | Error rate (%) | |
|---------|------|--------|-------|-------|-------|-------|-------|
| | | | Equal | Within twofold dilution | Within 4-fold dilution | Very major error | Major error |
| *C. neoformans* (*n* = 44) | | | | | | | |
| | AMB | BMD vs SYO | 25(57) | 42(95) | 43(98) | 10 (23%) / 0[b] | 7 (16%) / 0[b] |
| | 5FC | BMD vs SYO | 16(36) | 42(95) | 44(100) | 0 | 1(2) |
| | FLC | BMD vs SYO | 7(16) | 33(75) | 42(95) | 0 | 5(11) |
| | VRC | BMD vs SYO | 15(34) | 36(82) | 42(95) | 0 | 0 |
| | ITC | BMD vs SYO | 4(9) | 20(45) | 28(64) | 0 | 2(5) |
| | PSC | BMD vs SYO | 14(32) | 40(91) | 44(100) | 0 | 0 |
| *C. gattii* (*n* = 3) | | | | | | | |
| | AMB | BMD vs SYO | 3(100) | 3(100) | 2(67) | 0 | 0 |
| | 5FC | BMD vs SYO | 0 | 2(67) | 3(100) | 1(33) | 0 |
| | FLC | BMD vs SYO | 2(67) | 3(100) | 3(100) | 0 | 0 |
| | VRC | BMD vs SYO | 2(67) | 3(100) | 3(100) | 0 | 0 |
| | ITC | BMD vs SYO | 1(33) | 3(100) | 3(100) | 0 | 0 |
| | PSC | BMD vs SYO | 2(67) | 3(100) | 3(100) | 0 | 0 |

[a]CLSI, Clinical and Laboratory Standards Institute; BMD, broth microdilution; SYO, Sensititre YeastOne; AMB, amphotericin B; 5FC, flucytosine; FLC, fluconazole; VRC, voriconazole; ITC, itraconazole; PSC, posaconazole.
[b]Very major error and major error rates to AMB were according to ECV defined by CLSI M59/Clinical breakpoint defined by European Committee on Antimicrobial Susceptibility Testing.

## DISCUSSION

Our study demonstrated the comparison of antifungal susceptibilities of *Cryptococcus* isolates between the SYO version YO10C and VITEK2 YS09 with the standard CLSI BMD method. The MIC distributions of the novel antifungal agent—MGX according to the CLSI BMD method were investigated for the included *Cryptococcus* isolates.

In a study including 154 clinical yeast isolates, among which 16 isolates were *C. neoformans*, the essential agreement between SYO and VITEK2 with CLSI BMD was both >95%. Between SYO and CLSI BMD, there was one VME for FLC and one ME for VRC (22). In another study including 106 *C. neoformans* isolates from December 2010 to March 2018, the essential agreement between SYO and CLSI BMD was ≥93% for AMB, 5FC, FLC, and VRC. The VME was ≤6% while ME ≤2% between the two methods (21). In our study, the VME and ME for AMB reached 23% and 16%, respectively, between SYO and CLSI BMD according to the ECV defined by CLSI M59 (35), but no VME or ME was observed according to the clinical breakpoint defined by EUCAST (36).

**TABLE 6** Agreements and errors of 47 *Cryptococcus* isolates between the CLSI broth microdilution method and VITEK2 Susceptibility Card[a]

| Species | Drug | Method | Agreement (%) | | | Error rate (%) | |
|---------|------|--------|-------|-------|-------|-------|-------|
| | | | Equal | Within twofold dilution | Within 4-fold dilution | Very major error | Major error |
| *C. neoformans* (*n* = 44) | | | | | | | |
| | 5FC | BMD vs VITEK2 | 0 | 11 (25) | 27 (61) | 0 | 0 |
| | FLC | BMD vs VITEK2 | 7 (16) | 37 (84) | 43 (98) | 0 | 0 |
| | VRC | BMD vs VITEK2 | –[b] | –[b] | –[b] | 0 | 0 |
| *C. gattii* (*n* = 3) | | | | | | | |
| | 5FC | BMD vs VITEK2 | 0 | 0 | 0 | 1 (33) | 0 |
| | FLC | BMD vs VITEK2 | 0 | 0 | 2 (67) | 0 | 0 |
| | VRC | BMD vs VITEK2 | –[b] | –[b] | –[b] | 0 | 0 |

[a]CLSI, Clinical and Laboratory Standards Institute; BMD, broth microdilution; 5FC, flucytosine; FLC, fluconazole; VRC, voriconazole.
[b]The agreements were not applicable for VRC between two methods due to MIC of all Cryptococcus isolates ≤ 0.12 µg/mL except an undetermined result in one *C. neoformans* isolate.

In our study, the ME of FLC reached 11% between SYO and CLSI BMD according to the ECV defined by CLSI M59 (35). Discordant results were observed between previous studies. In a retrospective study, non-WT to FLC was found in 3% of clinical *C. neoformans* isolates, and the ME between SYO and CLSI BMD was 2% (21), but the study did not mention about the version of SYO used for antifungal testing. However, another retrospective study disclosed that the correlation of FLC MIC in *C. neoformans* isolates was weak between SYO and CLSI BMD method (40). Approximately 1.6% of clinical isolates had FLC MIC ≥ 16 µg/mL according to CLSI BMD, but the strains with FLC MIC ≥ 16 µg/mL increased to 53.2% by SYO. The version of SYO may significantly influence the ME between SYO and CLSI BMD, highlighting the rationale of our study.

In a recent retrospective study, the antifungal susceptibility reports by SYO did not predict clinical outcomes in 85 patients with *C. neoformans var. grubii* fungemia (23). In another retrospective study, the early and late mortality rates were not significantly different between WT and non-WT *Cryptococcus* species in patients with cryptococcal meningitis (24). The antifungal susceptibility patterns of clinical *Cryptococcus* isolates by SYO did not correlate with therapeutic outcomes. According to the CLSI BMD method, the *C. neoformans* isolates with MICs to FLC > 8 µg/mL was an independent predictive factor of poor clinical outcome in patients with cryptococcal meningitis (41). In another retrospective study, therapeutic failure was observed in patients who were infected with isolates for which fluconazole MICs were ≥16 µg/mL according to the CLSI BMD method (42). Our study investigated the performance of antifungal susceptibilities between SYO and CLSI BMD, but further studies about the correlation between different antifungal susceptibility testing methods and clinical outcomes in patients with cryptococcosis are required.

In most previous studies, there were no detailed recordings about the version of SYO used for antifungal susceptibility testing (18–22), so the antifungal susceptibility results for *Cryptococcus* isolates should be interpreted cautiously for potential off-label use, especially the *Cryptococcus* species was no more claimed by the manufacturer for susceptibility testing in the SYO version YO10C (26).

Despite limitation to FLC was claimed by manufacturer of VITEK2, the essential agreement among 62 *C. neoformans* isolates in India between VITEK2 and CLSI BMD was 82% for FLC. There was no VME or ME of FLC between the two methods (43). Similar results were found in our study with the essential agreement of *C. neoformans* isolates for FLC was approximately 98% between VITEK2 and CLSI BMD, and the agreement of MIC to FLC within twofold dilution was up to 84%. Although the essential agreement for 5FC was 61% between the two methods, the MIC distributions of 5FC with CLSI BMD were ranged 2–8 µg/mL while VITEK2 ranged ≤1–2 µg/mL.

The *in vitro* resistance of clinical *C. neoformans* isolates to antifungals was uncommon before 2000 (9). The incidence of *C. neoformans* with resistance to AMB, 5FC, or FLC remained low in a global surveillance study from 1990 to 2004 (10). Similar results were observed for even decreased trends of resistance of *C. neoformans* to FLC, PSC, and VRC in another global surveillance program from 1996 to 2008 (44). Possible explanations of the trends were possible influenced by the introduction of highly active antiretroviral therapy for human immunodeficiency virus (HIV) infected individuals and decreased incidence of cryptococcosis during the periods.

However, a surveillance program noticed that only 75% of *C. neoformans* isolates in North America displayed MIC to FLC ≤8 µg/mL, compared to 94%–100% in Africa, Europe, Latin America, and the Pacific regions during 1990–2004 (10). FLC MIC ≥64 µg/mL was observed in 6.2% environmental *C. neoformans* isolates obtained from 2006 to 2008 in Brazil (8). Besides, the *C. neoformans* isolates from HIV patients in Uganda presented elevated FLC MIC during 2010–2014 compared with previous studies during 1998–1999 (11). In a systemic review, the mean prevalence of FLC resistance was 12.1% of 4,995 *Cryptococcus* isolates from 3,210 patients although the definition of FLC resistance was different between studies (12).

A retrospective study included 59 hospitalized patients with cryptococcal infections at National Taiwan University Hospital during 1982–1997, among which 38 cases infected by *C. neoformans* while 21 caused by *C. gattii*. Isolates of *C. gattii* were found to be less susceptible to AMB and 5FC compared to *C. neoformans* (45). A nationwide study including 219 patients with proven cryptococcosis from 20 hospitals representative of all geographic areas in Taiwan during 1997–2010 found that antifungal MIC > ECV in 9 *C. neoformans* isolates, among which 7 to AMB, 1 to 5FC, and 1 to FLC. No MIC > ECV was observed in the *C. gattii* isolates (46). Among the 109 *C. neoformans* isolates collected from two medical centers in southern Taiwan from 2013 to 2020, one isolate had the MIC > ECV to FLC, while another one had MIC > ECV to PSC (47). The MIC < ECV was observed in six *C. gattii* isolates to AMB, 5FL, FLC, VRC, and PSC.

In our study, the AMB MIC of 11 *C. neoformans* isolates located at 1 µg/mL with the CLSI BMD method, which was above than the ECV (0.5 µg/mL) defined by CLSI M59 (35) but located at the upper limit of clinical breakpoint (1 µg/mL) by EUCAST (36). There was no isolate with MIC > ECV to FLC, VRC, ITC, PSA, or ISC with the CLSI BMD method in 47 *Cryptococcus* isolates in our study.

In previous *in vitro* studies of MGX against fungal infections, the MGX or its prodrug of fosmanogepix demonstrated the potency against yeasts and molds (28, 29). For MGX against 2,669 fungal isolates in the SENTRY Surveillance Program, the activity of MGX was proved with its MIC$_{50}$ 0.008 µg/mL for *Candida* species, 0.5 µg/mL for *C. neoformans* variant *grubii*, and 0.015 µg/mL for *Aspergillus* species isolates (48). However, the MGX MIC ≥ 1 µg/mL were observed in 93% ($n$ = 41) *C. neoformans* isolates and 100% ($n$ = 3) in *C. gattii* isolates in our study. Further investigations about susceptibilities to MGX of *Cryptococcus* isolates from different geographic areas are necessary.

The ERG11 gene alteration was considered to be associated with elevated MIC to FLC in *Cryptococcus isolates* (14–16). In a recent study, however, there was lacking of association between FLC susceptibilities and ERG11 nucleotide polymorphisms in clinical *C. neoformans* isolates from Uganda. The FLC MIC <8 µg/mL was observed in the isolates with ERG11 alterations, and the isolates with higher MIC may not contain ERG11 changes (49). Similar results were found in our study with the ERG11 mutation observed in up to 89% of *C. neoformans* isolates, and the mutation region was identical at I199V of all isolates. However, alterations of ERG11 gene were not associated with a decrease of susceptibility to FLC.

High genetic diversity with geographic differences was observed between *C. neoformans* isolates globally. The main ST of *C. neoformans* isolates in Brazil was ST 93, and no specific antifungal non-susceptibility was observed (50, 51). Approximately 50% of *C. neoformans* isolates in Iran was ST 77, and 53.3% of isolates was non-WT to AMB. However, there was no correlation between the ST and non-WT phenotypes to AMB (52). Over 75% of *Cryptococcus* isolates belonged to ST 31 in a study in India, and no association was observed between the ST and antifungal susceptibilities (53). In Japan, 82% of *C. neoformans* isolates was identified as ST 46 in an epidemiological study (54). In our study, up to 86% of *C. neoformans* isolates was classified as ST 5, and the ST was not correlated with elevated MIC to specific antifungal regimens. The results were consistent with a MLST-based genetic analysis of cryptococcus isolates in southern Taiwan from 2013 to 2020, in which the most common ST was type 5, and only one *C. neoformans* isolate had FLC MIC > 8 µg/mL (47).

There were several limitations in our study. First, the *Cryptococcus* isolates were collected in a single medical center in southern Taiwan, so the antifungal susceptibility distributions were not representative or applicable for those clinical isolates in other areas in Taiwan or abroad. Second, despite the collection of *Cryptococcus* isolates extended across for 10 years, the sample size was small. However, we noticed the ME to FLC with SYO compared to CLSI BMD in clinical isolates, so the results of SYO should be interpreted cautiously. Third, evaluations for therapeutic responses in patients with different antifungal susceptibilities of *Cryptococcus* isolates were difficult due to the small sample size, but our results showed the discrepant results between the SYO and CLSI

BMD methods. Despite the recommendation against using VITEK2 for determining FLC susceptibility, there were no VME or ME between VITEK2 and CLSI BMD method for FLC. Further investigations to compare the performance of different methods are required.

## Conclusion

The ME of FLC by SYO version YO10C compared to CLSI BMD reached up to 11% in clinical *C. neoformans* isolates in our study. The results of FLC MIC by SYO should be interpreted cautiously and correlated with therapeutic response, and further verification with the CLSI BMD method or VITEK2 is required.

## ACKNOWLEDGMENTS

We appreciate the technical assistances from the Diagnostic Microbiology and Antimicrobial Resistance Laboratory, National Cheng Kung University Hospital.

This study was supported by National Cheng Kung University Hospital Research Project (NCKUH-11204025, NCKUH-11301006, and NCKUH-11302061) and was funded by National Science and Technology Council, Taiwan (NSTC 113-2321-B-006-007 and NSTC 113-2314-B-006-045).

P.L.C. and T.P.W. conceived the study. S.W.W., S.L.S., and P.F.T. provided data collection. S.W.W., S.T.L., S.L.S., M.I.H., P.J.T., and P.F.T. provided technical assistances. P.L.C., T.P.W., C.J.W., N.Y.L., and W.C.K. analyzed the data and prepared the manuscript. All authors reviewed and edited the manuscript.

## AUTHOR AFFILIATIONS

[1]Center for Infection Control, National Cheng Kung University Hospital, College of Medicine, National Cheng Kung University, Tainan, Taiwan

[2]Department of Internal Medicine, National Cheng Kung University Hospital, College of Medicine, National Cheng Kung University, Tainan, Taiwan

[3]Department of Pathology, National Cheng Kung University Hospital, College of Medicine, National Cheng Kung University, Tainan, Taiwan

[4]Department of Medical Laboratory Science and Biotechnology, College of Medicine, National Cheng Kung University, Tainan, Taiwan

[5]Diagnostic Microbiology and Antimicrobial Resistance Laboratory, National Cheng Kung University Hospital, College of Medicine, National Cheng Kung University, Tainan, Taiwan

[6]National Institute of Infectious Diseases and Vaccinology, National Health Research Institutes, Tainan, Taiwan

[7]Department of Medicine, College of Medicine, National Cheng Kung University, Tainan, Taiwan

## AUTHOR ORCIDs

Tzu-Ping Weng  http://orcid.org/0009-0000-6266-1793
Po-Lin Chen  http://orcid.org/0000-0002-4015-1052

## AUTHOR CONTRIBUTIONS

Tzu-Ping Weng, Conceptualization, Data curation, Formal analysis, Investigation, Methodology, Project administration, Validation, Writing – original draft, Writing – review and editing | Shin-Wei Wang, Investigation, Methodology, Validation | Shih-Ting Lo, Investigation, Methodology | Shu-Li Su, Investigation, Methodology, Project administration | Ming-I Hsieh, Investigation, Methodology | Pei-Jane Tsai, Investigation, Methodology, Project administration | Pei-Fang Tsai, Data curation, Investigation, Methodology, Validation | Chi-Jung Wu, Investigation, Methodology, Validation | Nan-Yao Lee, Supervision, Validation, Writing – review and editing | Wen-Chien Ko, Supervision, Validation, Writing – review and editing | Po-Lin Chen, Conceptualization, Data curation, Formal analysis, Supervision, Validation, Writing – review and editing

## ADDITIONAL FILES

The following material is available online.

### Supplemental Material

**Supplemental material (Spectrum02117-24-s0001.pdf).** Tables S1 to S3.

### Open Peer Review

**PEER REVIEW HISTORY (review-history.pdf).** An accounting of the reviewer comments and feedback.

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
