## [Reviewer comments · Microbiology Spectrum]

Microbiology Spectrum

Comparative Evaluation of Sensitre YeastOne and VITEK2 Antifungal Susceptibility Tests with CLSI Broth Microdilution Method of Clinical *Cryptococcus* Isolates in Taiwan

Tzu-Ping Weng, Shin-Wei Wang, Shih-Ting Lo, Shu-Li Su, Ming-I Hsieh, Pei-Jane Tsai, Pei-Fang Tsai, Chi-Jung Wu, Nan-Yao Lee, Wen-Chien Ko, and Po-Lin Chen

Corresponding Author(s): Po-Lin Chen, National Cheng Kung University Hospital

Review Timeline:

Submission Date:	September 10, 2024
Editorial Decision:	November 7, 2024
Revision Received:	November 18, 2024
Accepted:	November 19, 2024

Editor: Po-Yu Liu

Reviewer(s): The reviewers have opted to remain anonymous.

Transaction Report:

DOI: <https://doi.org/10.1128/spectrum.02117-24>

Re: Spectrum02117-24 (Comparative Evaluation of Sensititre YeastOne and VITEK2 Antifungal Susceptibility Tests with CLSI Broth Microdilution Method of Clinical Cryptococcus Isolates in Taiwan)

Dear Prof. Po-Lin Chen:

Thank you for the privilege of reviewing your work. Below you will find my comments, instructions from the Spectrum editorial office, and the reviewer comments.

Revision Guidelines

Sincerely,
Po-Yu Liu
Editor
Microbiology Spectrum

Reviewer #1 (Comments for the Author):

This study assesses the reliability of commercial antifungal susceptibility tests (Sensititre YeastOne YO10C and VITEK2 YS09) compared to the CLSI broth microdilution (BMD) method in *Cryptococcus* isolates. Among 47 clinical isolates, findings indicate discrepancies in antifungal susceptibility, particularly with fluconazole (FLC) by SYO. The study emphasizes the need for cautious interpretation of SYO results and suggests confirmatory testing with CLSI BMD or VITEK2 for accurate therapeutic guidance.

Is there any possible explanation for the ME of FLC by SYO version YO10C reaching up to 11% in *C. neoformans* isolates, as reported in this study? Previous research (Comparison of MIC Test Strip and Sensititre YeastOne with the CLSI and EUCAST Broth Microdilution Reference Methods for In Vitro Antifungal Susceptibility Testing of *Cryptococcus neoformans*. *Antimicrob Agents Chemother* 64:10.1128/aac.02261-19) did not report such a high rate of major errors.

Reviewer #2 (Comments for the Author):

The authors have addressed all previous reviewer comments. The study does provide information that is valuable regarding off-label use of SYO version YO10C, which is surely to occur. This publication brings caution to that and should be published. The significance is well-reasoned and supported by the results.

Reviewer #3 (Comments for the Author):

In this study, the authors compared the performance of two commercially available antifungal susceptibility tests to the standard broth microdilution method for a set of 44 *Cryptococcal* clinical isolates from one hospital in Taiwan. One of the commercial tests, Sensititre YeastOne, was recently updated to exclude *C. neoformans* and the current study underscores the limitation of using this test for an unrecommended fungus as it reports major error rates ranging from 2-11%. The other commercial test, VITEK2, performed better but does not test as many antifungals.

The authors found no correlation between antifungal susceptibility and either sequence type or ERG11 mutation. They also used the CLSI BMD method to assess susceptibility for a promising new antifungal, MGX. For their set of clinical isolates, MIC of MGX ranged from 0.12-8 µg/ml, with 41 out of 44 above the MIC of 0.5 µg/ml reported for *C. neoformans* by the SENTRY Surveillance Program.

The study was conducted and analyzed well and the major findings are discussed adequately. I have one major point regarding data presentation and analysis and a few minor comments, listed below, that should be addressed in a revised manuscript.

Major comment:

In addition to the format in which the data is presented in Tables 1 and 2, I recommend also reporting the data reformatted by isolate (please see example Tables provided in attachment) as the additional insights gained by analyzing the data in this format will help offset the relatively small sample size. This analysis will reveal if the SYO method is failing to distinguish WT from non-WT for multiple antifungals (ie. are the 5 isolates non-WT to FLC by SYO among the 8 that are non-WT to AMB)? Additionally, it will be informative in determining if the 11 isolates that are non-WT to AMB by BMD also have high MIC for MGX.

Minor comments:

- 1) The authors cite recent reports that antifungal susceptibilities by SYO are poor predictors of clinical outcomes. Can the authors comment on how clinical outcomes correlate with the CSLI BMD method if this data is available?
- 2) Line 350 states that the MIC to ISC was less than or equal to 0.5 µg/mL determined by BMD and SYO but the data for SYO does not appear in Table 1.
- 3) The number of *C. neoformans* isolates with an AMB MIC of 1 µg/mL with the CLSI BMD method (Lines 475-476) should be 11 instead of 8.

In this study, the authors compared the performance of two commercially available antifungal susceptibility tests to the standard broth microdilution method for a set of 44 *Cryptococcal* clinical isolates from one hospital in Taiwan. One of the commercial tests, Sensititre YeastOne, was recently updated to exclude *C. neoformans* and the current study underscores the limitation of using this test for an unrecommended fungus as it reports major error rates ranging from 2-11%. The other commercial test, VITEK2, performed better but does not test as many antifungals.

The authors found no correlation between antifungal susceptibility and either sequence type or ERG11 mutation. They also used the CLSI BMD method to assess susceptibility for a promising new antifungal, MGX. For their set of clinical isolates, MIC of MGX ranged from 0.12-8 µg/ml, with 41 out of 44 above the MIC of 0.5 µg/ml reported for *C. neoformans* by the SENTRY Surveillance Program.

The study was conducted and analyzed well and the major findings are discussed adequately. I have one major point regarding data presentation and analysis and a few minor comments, listed below, that should be addressed in a revised manuscript.

Major comment:

In addition to the format in which the data is presented in Tables 1 and 2, I recommend also reporting the data reformatted by isolate (please see example Tables provided) as the additional insights gained by analyzing the data in this format will help offset the relatively small sample size. This analysis will reveal if the SYO method is failing to distinguish WT from non-WT for multiple antifungals (ie. are the 5 isolates non-WT to FLC by SYO among the 8 that are non-WT to AMB)? Additionally, it will be informative in determining if the 11 isolates that are non-WT to AMB by BMD also have high MIC for MGX.

Minor comments:

- 1) The authors cite recent reports that antifungal susceptibilities by SYO are poor predictors of clinical outcomes. Can the authors comment on how clinical outcomes correlate with the CSLI BMD method if this data is available?
- 2) Line 350 states that the MIC to ISC was less than or equal to 0.5 µg/mL determined by BMD and SYO but the data for SYO does not appear in Table 1.
- 3) The number of *C. neoformans* isolates with an AMB MIC of 1 µg/mL with the CLSI BMD method (Lines 475-476) should be 11 instead of 8.

Table 5 *In vitro* antifungal susceptibilities of 44 *Cryptococcus neoformans* isolates determined by CLSI broth microdilution method.

Isolate	Antifungal susceptibilities (MIC), µg/ml							
	AMB	5FC	FLC	VRC	ITC	PSC	ISC	MGX
1	0.5	4	4	0.12	<0.03	0.12	0.03	2
2	0.5	4	4	0.06	0.03	0.12	0.06	2
3	1	8	8	0.25	0.06	0.25	0.12	8
4	0.5	4	4	0.12	0.06	0.06	0.06	1
5	1	8	4	0.12	0.12	0.25	0.06	4
6	0.5	8	4	0.06	<0.03	0.12	0.03	2
7	0.5	4	4	0.06	0.06	0.12	0.06	2
8	0.5	2	2	0.06	<0.03	0.12	0.06	0.12
etc...

Non-WT (CLSI) indicated in red.

Table 6 *In vitro* antifungal susceptibilities of 44 *Cryptococcus neoformans* isolates determined by Sensititre YeastOne.

Isolate	Antifungal susceptibilities (MIC), µg/ml						
	AMB	5FC	FLC	VRC	ITC	PSC	ISC
1	0.5	4	8	0.06	0.25	0.25	0.06
2	1	16	16	0.25	0.5	0.5	0.25
3	0.5	4	8	0.12	0.12	0.25	0.12
4	0.5	4	8	0.06	0.12	0.25	0.06
5	0.5	2	8	0.12	0.12	0.25	0.12
6	0.25	4	4	0.06	0.03	0.12	0.06
7	0.5	8	8	0.06	0.12	0.25	0.12
8	0.5	4	8	0.12	0.25	0.12	0.12
etc...

Non-WT (CLSI) indicated in red.

Dear editor:

We sincerely appreciate your consideration and helpful suggestions regarding our manuscript entitled “**Comparative Evaluation of Sensititre YeastOne and VITEK2 Antifungal Susceptibility Tests with CLSI Broth Microdilution Method of Clinical *Cryptococcus* Isolates in Taiwan**” (Subject ID: Spectrum02117-24). We deeply appreciate the thorough analysis and comments by the reviewers. Each of your concerns has been clarified and addressed as well as incorporated into our revised manuscript. The following details are modifications in response to reviewers’ suggestions. The changes have been marked with underlines in the manuscript for your convenience to check all the changes we made in the revised manuscript.

Reviewer #1 (Comments for the Author):

This study assesses the reliability of commercial antifungal susceptibility tests (Sensititre YeastOne YO10C and VITEK2 YS09) compared to the CLSI broth microdilution (BMD) method in *Cryptococcus* isolates. Among 47 clinical isolates, findings indicate discrepancies in antifungal susceptibility, particularly with fluconazole (FLC) by SYO. The study emphasizes the need for cautious interpretation of SYO results and suggests confirmatory testing with CLSI BMD or VITEK2 for accurate therapeutic guidance.

Is there any possible explanation for the ME of FLC by SYO version YO10C reaching up to 11% in *C. neoformans* isolates, as reported in this study? Previous research (Comparison of MIC Test Strip and Sensititre YeastOne with the CLSI and EUCAST Broth Microdilution Reference Methods for In Vitro Antifungal Susceptibility Testing of *Cryptococcus neoformans*. *Antimicrob Agents Chemother* 64:10.1128/aac.02261-19) did not report such a high rate of major errors.

Thank you for reviewer’s question to investigate the discrepancy of major error (ME) rates of fluconazole (FLC) by Sensititre YeastOne (SYO) version YO10C and CLSI broth microdilution method (BMD) method in clinical *C. neoformans* isolates

between our study and another retrospective study¹. For susceptibilities to FLC of *C. neoformans* isolates in our study, there was no non-wild type (WT) strain to FLC observed according to the CLSI BMD method, which was consistent with previous studies in Taiwan^{2,3} and abroad^{4,5}. However, the ME of FLC between SYO version YO10C and CLSI BMD reached 11% in our study.

In the retrospective study mentioned by reviewer¹, non-WT to FLC was found in 3% of clinical *C. neoformans* isolates, and the ME between SYO and CLSI BMD was 2%. However, the study didn't mention about the version of SYO used for antifungal testing in methodology section. Besides, another study cited by this study disclosed that the correlation of FLC MIC of *C. neoformans* isolates was weak between SYO and CLSI BMD method⁶. Approximately 1.6% of clinical isolates had FLC MIC ≥ 16 $\mu\text{g/mL}$ according to CLSI BMD, but the strains with FLC MIC ≥ 16 $\mu\text{g/mL}$ increased to 53.2% by SYO. The version of SYO may significantly influence the ME between SYO and CLSI BMD, highlighting the rationale of our study. Thank you for giving us an opportunity to explain the issue in detail. The discussion of the issue has been supplemented in the manuscript. (Line 421-431)

Reviewer #2 (Comments for the Author):

The authors have addressed all previous reviewer comments. The study does provide information that is valuable regarding off-label use of SYO version YO010C, which is surely to occur. This publication brings caution to that and should be published. The significance is well-reasoned and supported by the results.

Thank you for reviewer's attention and comment for the manuscript of our study.

Reviewer #3 (Comments for the Author):

In this study, the authors compared the performance of two commercially available antifungal susceptibility tests to the standard broth microdilution method for a set of 44 Cryptococcal clinical isolates from one hospital in Taiwan. One of the commercial tests, Sensititre YeastOne, was recently updated to exclude *C. neoformans* and the current study underscores the limitation of using this test for an unrecommended fungus as it reports major error rates ranging

from 2-11%. The other commercial test, VITEK2, performed better but does not test as many antifungals.

The authors found no correlation between antifungal susceptibility and either sequence type or ERG11 mutation. They also used the CLSI BMD method to assess susceptibility for a promising new antifungal, MGX. For their set of clinical isolates, MIC of MGX ranged from 0.12-8 µg/ml, with 41 out of 44 above the MIC of 0.5 µg/ml reported for *C. neoformans* by the SENTRY Surveillance Program.

The study was conducted and analyzed well and the major findings are discussed adequately. I have one major point regarding data presentation and analysis and a few minor comments, listed below, that should be addressed in a revised manuscript.

Major comment:

In addition to the format in which the data is presented in Tables 1 and 2, I recommend also reporting the data reformatted by isolate (please see example Tables provided in attachment) as the additional insights gained by analyzing the data in this format will help offset the relatively small sample size. This analysis will reveal if the SYO method is failing to distinguish WT from non-WT for multiple antifungals (ie. are the 5 isolates non-WT to FLC by SYO among the 8 that are non-WT to AMB)? Additionally, it will be informative in determining if the 11 isolates that are non-WT to AMB by BMD also have high MIC for MGX.

I appreciate the reviewer's recommendation to format the tables with antifungal MIC distributions of each *C. neoformans* isolate. According to non-WT *C. neoformans* isolates observed in CLSI BMD method and SYO (Table 1 in the manuscript), I produced the Table 5 and Table 6 for antifungal susceptibilities reports of *C. neoformans* isolates by the two methods, respectively.

Among the 11 isolates belonged to non-WT to AMB by CLSI BMD method, the measured MIC to MGX ranged from 0.25 µg/ml to 4 µg/ml (Table 5). Among the 5 isolates belonged to non-WT to FLC by SYO, there was only one isolate classified as non-WT to AMB (Isolate number 28 in Table 6). Thank you for reviewer's suggestion to help us clarifying the two issues. The findings and the tables have been supplemented in the manuscript (Line 339, 347-348, 867-893)

Minor comments:

1) The authors cite recent reports that antifungal susceptibilities by SYO are poor predictors of clinical outcomes. Can the authors comment on how clinical outcomes correlate with the CLSI BMD method if this data is available?

Thank you for the reviewer's question about the issue. According to the two studies^{7,8}, there was lacking of association between antifungal susceptibilities by SYO and clinical outcomes in patients with cryptococcal meningitis and fungemia. However, the methodology of the two studies didn't include CLSI BMD method for comparison, so the association between CLSI BMD and clinical outcomes in patients with cryptococcosis was not available.

According to a previous study, the *C. neoformans* isolates with MIC >8 µg/ml according to CLSI BMD method was an independent predictive factor of poor clinical outcome in patients with cryptococcal meningitis⁹. In another prior retrospective study, therapeutic failure was observed in patients who were infected with isolates for which fluconazole MICs were ≥16 µg/ml according to the CLSI BMD method¹⁰.

Our study investigated the performance of SYO compared with CLSI BMD method, but further studies about correlation between different antifungal susceptibility testing methods and clinical outcomes in patients with cryptococcosis are required. Thank you for providing me the opportunity to discuss the issue in detail, and the issue was supplemented in the discussion section of the manuscript (Line 439-446)

2) Line 350 states that the MIC to ISC was less than or equal to 0.5 µg/mL determined by BMD and SYO but the data for SYO does not appear in Table 1. I sincerely appreciate reviewer's attention to the mistake and inadequate of wording. The sentence should be separated and specified to the BMD and SYO, respectively. Isavuconazole (ISC) MIC testing is not available for SYO, which had been recorded in the methodology section of the manuscript (Line 245-251). I have separated the sentence to "the MICs to ITC and PSC were all ≤ 0.5 µg/mL determined by BMD and SYO, while the MICs to ISC were all ≤ 0.5 µg/mL by BMD". (Line 351-353)

3) The number of *C. neoformans* isolates with an AMB MIC of 1 µg/mL with the CLSI BMD method (Lines 475-476) should be 11 instead of 8.

Thank you very much for pointing out the mistake in the manuscript. I have corrected the number of *C. neoformans* isolates with an AMB MIC of 1 µg/mL by the CLSI BMD method from 8 to “11”. Thanks for reviewer’s reminder for the mistake. (Line 496-497)

I have corrected another mistake in Table 1 that the number of *C. neoformans* isolates with FLC MIC of 4 µg/mL was “28” instead of 29, and the isolates with FLC MIC of 8 µg/mL was “6” instead of 5. The mistake has been corrected in Table 1. (Line 816)

We sincerely appreciate the reviewers’ attention and notifications to the mistakes and inadequacies of the manuscript. The manuscript has been corrected and renewed, and the questions were replied point-by-point. I appreciate the reviewers’ comments and make detailed modifications to optimize the contents of the manuscript.

Tzu-Ping Weng, MD

Department of Internal Medicine, National Cheng Kung University Hospital, No. 138, Sheng Li Road, 704, Tainan, Taiwan

Email: henrywon12382@yahoo.com.tw

***Corresponding author:**

Po-Lin Chen, MD

Department of Internal Medicine, National Cheng Kung University Hospital, No. 138, Sheng Li Road, 704, Tainan, Taiwan

Email: cplin@mail.ncku.edu.tw

Reference:

1. Delma FZ, Al-Hatmi AMS, Buil JB, van der Lee H, Tehupeiori-Kooreman M, de Hoog GS, Meletiadis J, Verweij PE. Comparison of MIC Test Strip and Sensititre YeastOne with the CLSI and EUCAST Broth Microdilution Reference Methods for *In Vitro* Antifungal Susceptibility Testing of *Cryptococcus neoformans*. *Antimicrob Agents Chemother.* 2020 Mar 24;64(4):e02261-19.
2. Tseng HK, Liu CP, Ho MW, Lu PL, Lo HJ, Lin YH, Cho WL, Chen YC; Taiwan Infectious Diseases Study Network for Cryptococcosis. 2013. *Microbiological,*

epidemiological, and clinical characteristics and outcomes of patients with cryptococcosis in Taiwan, 1997-2010. PLoS One 8(4):e61921.

3. Chen YC, Kuo SF, Lin SY, Lin YS, Lee CH. 2022. Epidemiological and Clinical Characteristics, Antifungal Susceptibility, and MLST-Based Genetic Analysis of *Cryptococcus* Isolates in Southern Taiwan in 2013-2020. J Fungi (Basel) 8(3):287.
4. Pfaller MA, Messer SA, Boyken L, Rice C, Tendolkar S, Hollis RJ, Doern GV, Diekema DJ. 2005. Global trends in the antifungal susceptibility of *Cryptococcus neoformans* (1990 to 2004). J Clin Microbiol 43(5):2163-7.
5. Pfaller MA, Castanheira M, Diekema DJ, Messer SA, Jones RN. 2011. Wild-type MIC distributions and epidemiologic cutoff values for fluconazole, posaconazole, and voriconazole when testing *Cryptococcus neoformans* as determined by the CLSI broth microdilution method. Diagn Microbiol Infect Dis 71(3):252-9.
6. Vena A, Muñoz P, Guinea J, Escribano P, Peláez T, Valerio M, Bonache F, Gago S, Álvarez-Uría A, Bouza E. Fluconazole resistance is not a predictor of poor outcome in patients with cryptococcosis. Mycoses. 2019 May;62(5):441-449.
7. Wu TS, Lin JF, Cheng CW, Huang PY, Yang JH. Lack of Association between YEASTONE Antifungal Susceptibility Tests and Clinical Outcomes of *Cryptococcus* Meningitis. J Fungi (Basel). 2023 Feb 10;9(2):232.
8. Yang JH, Huang PY, Cheng CW, Shie SS, Lin ZF, Yang LY, Lee CH, Wu TS. Antifungal susceptibility testing with YeastONE™ is not predictive of clinical outcomes of *Cryptococcus neoformans* var. *grubii* fungemia. Med Mycol. 2021 Nov 3;59(11):1114-1121.
9. Lee CH, Chang TY, Liu JW, Chen FJ, Chien CC, Tang YF, Lu CH. Correlation of anti-fungal susceptibility with clinical outcomes in patients with cryptococcal meningitis. BMC Infect Dis. 2012 Dec 20;12:361.
10. Aller AI, Martin-Mazuelos E, Lozano F, Gomez-Mateos J, Steele-Moore L, Holloway WJ, Gutiérrez MJ, Recio FJ, Espinel-Ingroff A. Correlation of fluconazole MICs with clinical outcome in cryptococcal infection. Antimicrob Agents Chemother. 2000 Jun;44(6):1544-8.

Re: Spectrum02117-24R1 (Comparative Evaluation of Sensititre YeastOne and VITEK2 Antifungal Susceptibility Tests with CLSI Broth Microdilution Method of Clinical Cryptococcus Isolates in Taiwan)

Dear Prof. Po-Lin Chen:

Your manuscript has been accepted, and I am forwarding it to the ASM production staff for publication. Your paper will first be checked to make sure all elements meet the technical requirements. ASM staff will contact you if anything needs to be revised before copyediting and production can begin. Otherwise, you will be notified when your proofs are ready to be viewed.

Sincerely,
Po-Yu Liu
Editor
Microbiology Spectrum